# TOWARD FAITHFUL CASE-BASED REASONING THROUGH LEARNING PROTOTYPES IN A NEAREST NEIGHBOR-FRIENDLY SPACE.

**Omid Davoodi**
School of Computer Science
Carleton University
Ottawa, ON, Canada
`omid.davoudi@carleton.ca`

**Majid Komeili**
School of Computer Science
Carleton University
Ottawa, ON, Canada
`majid.komeili@carleton.ca`

## ABSTRACT

Recent advances in machine learning have brought opportunities for the ever-increasing use of AI in the real world. This has created concerns about the black-box nature of many of the most recent machine learning approaches. In this work, we propose an interpretable neural network that leverages metric and prototype learning for classification tasks. It encodes its own explanations and provides an improved case-based reasoning through learning prototypes in an embedding space learned by a probabilistic nearest neighbor rule. Through experiments, we demonstrated the effectiveness of the proposed method in both performance and the accuracy of the explanations provided.

## 1 INTRODUCTION

With the ever-increasing usage of deep learning in real-world situations, interpretable and explainable machine learning has become more and more important. This is because the deeply complex nature and large number of parameters in deep learning models raise concerns about reliability, fairness, transparency, and trust. While some research has been focused on explaining the current "black-box" models by offering extra insight about their decisions (Ribeiro et al., 2016; Lundberg & Lee, 2017; Selvaraju et al., 2017), another approach is to create models that are inherently interpretable (Rudin, 2019).

Some classes of machine learning models are deemed to be inherently more interpretable(Doshi-Velez & Kim, 2017). These include linear models like logistic regression, decision trees, rules-based models, naive bayes, and nearest neighbor models. These models are usually simple or abstract enough that a human can get an understanding of how they make specific decisions. Decisions made by nearest neighbor methods, in particular, can be easily understood by a human. But an important disadvantage of nearest neighbor models is that their inference computational cost scales with the amount of training data. This can make them impractical when the dataset is too large.

One of the main ways suggested to combat the cost of nearest neighbor models is to use prototypes. Prototypes are points in the dataset that represent a group of points around them. If the number of prototypes is smaller than the number of data points, using them as a proxy of real data for nearest neighbor methods could significantly speed up the inference process. Depending on the number and the position of prototypes, performance differences compared to using the whole dataset can be minimal.

Using prototypes for classification is not a new concept. Research was done on nearest prototype classifiers (Bezdek & Castelaz, 1977; Kohonen, 1990; Seo et al., 2003) and their validity (Kuncheva & Bezdek, 1998; Graf et al., 2009). One important domain where these types of classifiers are used is in interpretable classification (Bien & Tibshirani, 2011; Kim et al., 2014; Chen et al., 2018; Li et al., 2018). Recently Li et al. (2018)

presented an explainable neural network structure that represents samples by their distance to some prototypes learned in an embedding space. A fully connected layer gets these distances as input and outputs the final class. It learns the prototypes along with the network weights. However, the drawback of that method is that it learns the prototypes in a poorly constructed embedding space. Because the structural similarities in the embedding space are not preserved, prototypes are less likely to be surrounded by a group of similar samples. Since these prototypes are the explanations for the model, naturally one may expect that the prototypes should represent unknown homogenous clusters that might exist within classes (i.e. subclasses). For example different breeds of dogs within the class dog. We empirically show that Li et al. (2018) fails to preserve such subclasses and even destroys them. We argue that this is due to the lack of a mechanism for learning a proper embedding space in conjugation with learning the prototype positions.

In this paper, we offer an interpretable prototype classification method called Deep Embedded Prototype Network (DEPN). This method is a neural network architecture that finds a good embedding space and a number of suitable prototypes for classification. The embedding space and the prototypes in DEPN are constructed in such a way that high classification performance is achieved. In addition, the decisions of the classifier can be explained by showing the nearest prototypes as a comparative example. We show that the combination of our embedding space and prototype selection method leads to cohesive prototype explanations compared to the state-of-the-art Li et al. (2018).

The embedding space of DEPN is formed such that samples that are closer to each other in terms of Euclidean distance are more similar to each other than samples that are further away. It is because the embedding space is learned based on the nearest-neighbor rule. In particular, following (Chauhan et al., 2021) an upper bound of the classification error using nearest-neighbour classifier is minimized. While it is very hard to find annotated datasets containing similarity measures between each pair of samples, it is possible to create datasets that contain superclasses each consisting of multiple subclasses. Subclass similarity is, therefore, a good proxy for overall sample similarity. The advantage of (Chauhan et al., 2021) is that it has been shown to be able to create embedding spaces where even unlabeled subclasses of the data distribution are preserved in the embedding space. This happens despite the model not having any direct knowledge about subclass data. This means that for example, if one of the classes in the data consists of two subclasses, it is less likely to mix these two subclasses together in the resulting embedding space.

For example, suppose that the problem is the classification between cats and dogs using only a picture of the animal. If the dataset consists of multiple dog and cat breeds, we can assume that it contains multiple subclasses of cats and dogs. If the nearest neighbors of a dog sample in the embedding space are other dog samples that belong to the same breed (subclass), the closest samples to a prototype might also belong to the same subclass. We believe that if the subclasses of the closest prototype used to classify a new sample is the same as that of that sample, the prototype is a better explanation for the decision taken by the model compared to when the overall class label is correct, but the subclass shows a different breed of the animal. It is important to note that the method we provide does not have access to the subclass data. In fact, it doesn't need the subclass labels to be present at all.

The final result is a neural network that classifies the input using a number of prototypes. It can explain its decisions by offering the closest prototype of that class to the human user. Due to the way our embedding space preserves subclass information, the offered explanation is much more likely to be suitable compared to the previous state-of-the-art method Li et al. (2018). In particular, our measurements show that our work preserves the subclass neighborhoods in the embedding space while Li et al. (2018) actively degrades them. This is while DEPN has the same overall performance when measured in superclass accuracy while showing, on average, 125% improvement in subclass accuracy.

There are previous works that might seem similar to our own. Deep k-Nearest-Neighbors (DkNN) by Papernot & McDaniel (2018) is an interpretability-based method for deep neural networks. This method uses nearest-neighbor labels in the intermediate layers of a neural network to assess the validity of a decision. The main idea behind DkNN is that when the network makes a decision, the nearest neighbors in the intermediate layers should also share the same label. If this is not the case for most of the layers, then maybe this decision

is something that came up due to the quirks of the final layers of the network. While this focus on nearest neighbors might seem similar to our work, DkNN ultimately offers a confidence score for the decisions of the network. In contrast, our method offers examples (prototypes) as explanations for the decisions it takes. In fact, DkNN can be used alongside DEPN to offer confidence scores for its decisions.

The work done by Li et al. (2018) is another method that is quite similar to our work. While the network architecture in both works is mostly the same (especially if MsDNN-AE is used for the encoder), the way they utilize the architecture is quite different. This work is trained end-to-end, but DEPN uses a more gentle two-stage training procedure. The $r_1$ loss used in this work does not consider the prototype labels while the one in DEPN does. This work uses the decoder outputs as the representations of its prototypes while the decoder in DEPN is an optional regularization system for MsDNN. If the decoder is not able to output good representations, this work will not be able to offer proper explanations for its decisions. The explanations in DEPN in contrast, are the nearest same class prototype, which will be represented by a training sample. Another difference is that the explanations offered by this method are much less likely to be of the same subclass as that of DEPN.

The work done by Chen et al. (2018) offers prototype explanations-based on parts of an image. These explanations are inherently different from the sample based explanations DEPN offers. This method is also limited to data that can be split into parts and will not be feasible for other types of data (like tabular data).

## 2 METHODOLOGY

Our method is a neural network with an uncommon structure suited for prototype training. It consists of a backbone encoder network that transforms the input to a metric space learned based on a nearest-neighbor rule. There is an optional decoder that will be used in an autoencoder which will get its inputs from the embedding space and reconstruct the original input. Then, we have $n$ prototypes which will be modeled as a special layer on top of the encoder. The value of $n$ is a hyperparameter. The larger the $n$, the more detailed and specific our explanations will become at the expense of more computational cost. Finally, we have a normal fully connected neural network layer that gets the output of the prototype layer and outputs the class likelihoods. This output layer can have a softmax activation function.

The special prototype layer in our method gets the transformed samples from the embedding space as input and outputs a vector for each sample denoting the distance between that sample and each prototype in the embedding space. This arrangement was first proposed by Li et al. (2018) and lets us position the prototypes during the training process of the neural network itself. a schema of our network architecture is illustrated in Figure 2.

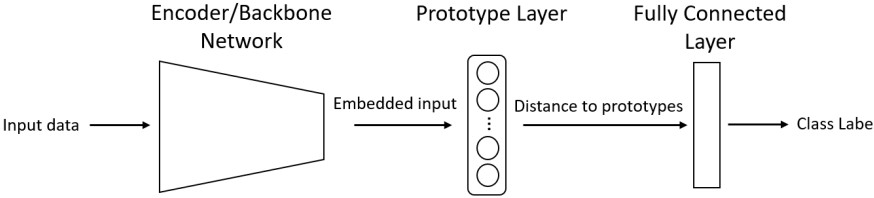

Figure 1: A schema of our network architecture

### 2.1 TRAINING PROCEDURE

The training happens in two steps: The first step is to train the embedding space so that a good embedding space is formed at the output of the encoder. This is done by using the special MsDNN loss (Chauhan et al.,

2021) on the embedding space. MsDNN loss is a variation of triplet loss (Weinberger & Saul, 2009) that uses Gaussian estimations of the overall positive and negative samples as a substitute of the positive and negative samples themselves. It has been shown to be effective in preserving the unseen subclasses of the data and consequently is a good pick for our purpose. Optionally, a decoder that gets the transformed embeddings as inputs and reconstructs the original samples can also be used in this stage of the training, similar to MsDNN-AE (Chauhan et al., 2021).

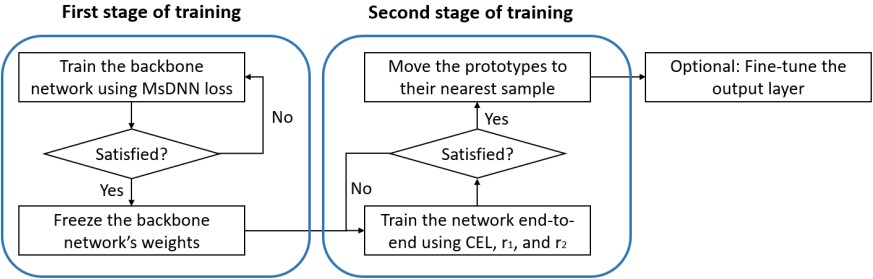

Figure 2: A schema of the training procedures for DEPN

Training stops when accuracy using nearest-neighbor classification on a validation set is maximized. Our experiments show that unseen subclass accuracy, when measured by nearest-neighbor classification accuracy, generally follows the coarse-grained classification. This lets us optimize our embedding space using only the coarse-grained class data and still get within the general vicinity of the best settings for fine-grained subclass accuracy in the embedding space without having any subclass labels.

The choice of hyperparameter $\sigma$ in MsDNN is important to the viability of the created embedding space. This hyperparameters determines the size of the effective area for weighting the samples. Higher values would result in further samples being considered close neighbors for creating the positive and negative data point for MsDNN's contrastive loss. We found that starting from small values and doubling the value of $\sigma$ until the peak of coarse-grained validation accuracy to be a good procedure. Extremely large values of $\sigma$ encourage subclasses to merge. We found out that fine-grained and coarse-grained accuracy follow the same pattern with respect to the value of $\sigma$. This means that coarse-grained validation accuracy is generally a good indicator of fine-grained accuracy as well.

The second step of the training procedure trains the whole network in an end-to-end manner for classification but with the weights of the encoder network frozen. In essence, what this stage does is to train the prototype and the output layers of the network. The input samples are fed to the encoder and the output of the output layer is used for the final classification. The loss function used in this stage can be seen in equation 1.

$$L = \lambda_1 CEL(\bar{y}, y) + \lambda_2 r_1(e, p) + \lambda_3 r_2(e, p) \tag{1}$$

Where $L$ is the loss value, $CEL$ is the cross-entropy loss, $\bar{y}$ represents the true coarse-grained labels of the input samples, $y$ is the predicted output of the network, $r_1$ and $r_2$ are special losses defined on the prototype layer, $e$ represents the transformed training samples in the embedding space, $p$ represents the set of learned prototypes in the embedding space, and $\lambda_1, \lambda_2, \lambda_3 \in \mathbb{R}^+$ are weights of each individual loss function. $r_1$ and $r_2$ are defined as:

$$r_1 = \sum_{e_i \in e} \min_{p_j \in p_{c_i}} (\|p_j - e_i\|_2^2), p_{c_i} = \{p_k \in p | \exists n : y_n = y_i, \|p_k - e_n\|_2 = \min_{e_m \in e}(\|p_k - e_m\|_2)\} \tag{2}$$

$$r_2 = \sum_{p_i \in p} \min_{e_j \in e}(\|p_i - e_j\|_2^2) \tag{3}$$

Table 1: Results of experiments on MNIST and FashionMNIST datasets.

| Methods | MNIST | | | |
| | Coarse | Fine | C-NN | F-NN |
|---|---|---|---|---|
| PDL | **98.39% ±0.51%** | 33.82% ±1.38% | 98.59% ±0.11% | 88.45% ±0.54% |
| DEPN-A | 97.81% ±0.95% | **88.36% ±1.13%** | **98.74% ±0.07%** | **96.65% ±0.31%** |
| DEPN | 92.79% ±0.99% | 83.64% ±1.52% | 98.53% ±0.09% | 94.89% ±0.42% |
| DEPN-I | 79.8% ±0.57% | 76.13% ±1.89% | 97.94% ±0.00% | 96.63% ±0.00% |
| Methods | Fashion-MNIST | | | |
| | Coarse | Fine | C-NN | F-NN |
| PDL | **94.03% ±0.24%** | 41.53% ±2.71% | 92.74% ±0.09% | 69.75% ±0.57% |
| DEPN-A | 92.38% ±0.51% | **67.14% ±1.43%** | 92.68% ±0.09% | **82.02% ±0.32%** |
| DEPN | 92.01% ±0.35% | 57.83% ±1.57% | **92.79% ±0.12%** | 78.57% ±0.23% |
| DEPN-I | 86.11% ±0.36% | 59.68% ±1.22% | 91.86% ±0.00% | 81.48% ±0.00% |

Where $e_i$ is an element in $e$ with index $i$, $p_{c_i}$ is the set of all prototypes that have the same class label as $e_i$ (determined by the the label of the closest sample in the current batch to that prototype) and $p_j$ is an element of $p_{c_i}$. $r_1$ is a loss function that sums up the distance between each input sample in the current batch and its nearest same class prototype. The class label of a prototype in this case is the class label of its nearest sample in the current batch. On the other hand, $r_2$ sums up the distance between each prototype and its closest input sample in the current batch. Minimizing $r_2$ will make sure that each prototype is close to at least one input sample. Minimizing $r_1$ on the other hand makes sure that each sample is close to at least one prototype of the same class. Minimizing these two losses combined makes sure that the prototypes are well-positioned inside the distribution of the data. To increase the speed of the process, the initial position of the prototypes will be chosen as the position of a number of randomly selected training samples.

Minimizing the overall loss function will position the prototypes in key positions of the data distribution and will train the output layer to use the distance of samples to each prototype to classify them. It is important to note that the freezing of the weights of the encoder ensures that these loss functions do not determine the embedding space itself and only use it to classify the samples using the prototypes. We found that not freezing the encoder will result in the destruction of the subclass neighborhoods in the embedding space.

The stopping criteria in this stage of training are the plateauing of the loss function and the validation accuracy. In our experience, the latter happens sooner than the former. At this stage, the prototypes should be manually moved to the nearest sample in the training set. This sample is called the *prototype sample* and its class label is used as the class label of that prototype. At this point, the network is ready for inference. An optional procedure after this is to fine-tune the output layer using only cross-entropy loss due to the mentioned moving of the prototypes, but we saw no noticeable change in the output of the network even without this fine-tuning. A schema of the training procedures can be found in Figure 2.1.

## 2.2 INFERENCE AND EXPLANATIONS

The inference phase of DEPN is relatively simple. First, the data points are fed to the network and are classified using the output layer. Then, the closest prototype to the embedding of that data point that shares the same class label as the one predicted by the network is selected. The prototype sample of that prototype is then offered as an explanation to the end-user.

## 3 EXPERIMENTS

To assess the validity of our method, we performed a series of experiments. Our first experiments were done on the MNIST (Deng, 2012) and FashionMNIST(Xiao et al., 2017) datasets. To evaluate the ability

of the networks to deal with unlabeled subclasses, we combined the first 5 classes of these datasets into one superclass. The same was done for the last 5 classes. This created a 2-way classification problem where we also have subclass labels to be used for evaluation purposes (not training). We compare our work with the one by Li et al. (2018) which is the state of the art in prototype-based explainable networks and from now on will be referred to as PDL (Prototype Deep Learning). In all experiments, the number of prototypes was 30. They were trained on 90% of the training set and validated on the remaining 10%. Adam optimizer (Kingma & Ba, 2014) was used in all instances. The results mentioned here are from evaluating on the test set. The results contain the coarse-grained (superclass) network accuracy indicated as *Coarse*, fine-grained (real classes) network accuracy by using the fine-grained label of the closest same-class prototype as the prediction of the network. It is indicated as *Fine*. We also calculated both coarse and fine-grained nearest-neighbor accuracy in the embedding spaces of both methods (C-NN and F-NN respectively) as additional measures of how good each embedding space is. Note that F-NN represents a classic nearest-neighbor classification result while Fine represents a nearest-prototype classification accuracy. All experiments were repeated 8 times and their results were averaged.

PDL was trained for 50 epochs on a backbone network that consisted of 4 convolutional layers. All layers except the last one had 32 feature maps. The last layer had 10 feature maps instead. The activation function for all of the backbone layers was ReLU. The decoder network was the mirror image of the encoder, using deconvolution layers instead. The weight for cross-entropy loss was set to 20 and the weights of MSE in the auto-encoder, $r_1$ and $r_2$ were set to 1 during training.

DEPN was trained for 50 epochs on the same backbone network at the first stage and then for another 30 epochs at the second stage. Experiments were done using both MsDNN and MsDNN-AE (DEPN and DEPN-A respectively). When MsDNN-AE was used, the decoder was also the same as the previous experiments. Best $\sigma$ for MsDNN-AE was found to be 0.128 and the weights of the second stage of training were 10 for cross-entropy loss and $r_1$ and 1 for $r_2$. An experiment was also done in which MsDNN was not used at all and the prototypes were instead placed in the input space itself with no backbone network (DEPN-I). Only the second stage of training was performed in this experiment. This was done as an ablation study on the use of MsDNN itself. The results for these experiments can be seen in Table 1

Another set of experiments were done on the CIFAR-10 dataset. Similar to MNIST, we created two super-classes, one containing planes, cars, cats, birds, and deer subclasses and another one containing dogs, frogs, horses, ships, and trucks. For this experiment, we used a pre-trained Resnet-18 (He et al., 2016) network as our backbone. As Resnet-18 uses input sizes of 224×244 pixels while CIFAR-10 images are only 32×32, we scaled the inputs to 224×224 using a linear interpolation method. The number of distinct dimensions in our input data was 32×32×3=3072 and the output of the resnet-18 has 2×2×128=512 dimensions. We wanted our embedding space to be smaller so as to force the network to compress the image data even more, so we added a new fully-connected layer with 120 neurons to the end of the resnet network. This layer had a LeakyReLU activation with a 0.2 scaling factor. The decoder used in this configuration had 6 initial de-convolutional layers with 64 feature maps and a LeakyReLU activation function with a 0.2 scaling factor. These layers fed into a final layer with 3 feature maps that represented the RGB values of the original image, recreating the input at its output. All of these layers had a kernel size of 4 and a stride of 2. All of them except the third layer had a padding of 1. The third layer used a padding of 2. This configuration made sure that our output size matched the 224×224 pixel image that was used as input. All of these experiments used a prototype count of 40.

For PDL, we used a weight distribution of 20 for cross-entropy loss and 1 for all of the remaining loss functions. This is similar to the original values used in Li et al. (2018). The network was trained for 150 epochs and evaluated the same way as the previous experiments. For DEPN, the best value for $\sigma$ was found to be 4 and the weights of the loss functions were set as 10 for both cross-entropy loss and $r_1$ and 1 for $r_2$. The reason $r_2$ had a lower weight was because the batch-size used was 200 and this would adversely affect $r_2$ itself. More explanations on $r_1$, $r_2$, and the value of $\sigma$ are provided in the appendix. The network trained for the first stage for 130 epochs and 50 for the second stage. These values ensured that all of our validation losses in all of the experiments plateau long enough for us to be sure about the fact that the training is done.

Table 2: Results of experiments on CIFAR-10.

| Methods | Coarse | Fine | C-NN | F-NN |
|---------|--------|------|------|------|
| PDL | 91.05% ±0.36% | 19.50% ±1.14% | 91.88% ±0.22% | 40.11% ±0.68% |
| DEPN-A | **91.12% ±0.76%** | **58.29% ±2.59%** | **92.45% ±0.65%** | **74.70% ±0.21%** |
| DEPN | 90.50% ±0.41% | 56.88% ±1.54% | 92.00% ±0.36% | 72.01% ±1.34% |
| DEPN-I | 59.41% ±0.24% | 36.38% ±3.60% | 80.27% ±0.00% | 70.07% ±0.00% |

Table 3: Results of experiments on ImageNet.

| Methods | Coarse | Fine | C-NN | F-NN |
|---------|--------|------|------|------|
| PDL | 96.61% ±0.22% | 33.37% ±1.46% | 96.28% ±0.31% | 42.81% ±1.18% |
| DEPN-A | **97.02% ±0.24%** | 54.17% ±3.64% | **97.14% ±0.34%** | 80.68% ±0.63% |
| DEPN | 96.19% ±0.10% | 60.45% ±7.90% | 97.00% ±0.47% | 82.63% ±1.72% |
| DEPN-I | 95.28% ±0.31% | **61.31% ±4.45%** | 95.66% ±0.00% | **85.75% ±0.00%** |

The evaluation scheme was the same as the previous experiments. The only exception was that in DEPN-I, the prototypes were placed in the Resnet-18 embedding space (and not in the space of the additional fully connected layer we added to the network). This was because Resnet's embedding space is pre-trained and could be used to determine how much of the performance was due to Resnet itself. The results can be seen in Table 2

The last set of experiments was done on a subset of the ImageNet dataset (Deng et al., 2009). The previous experiments combined unrelated classes of existing datasets into superclasses. With ImageNet, we have a hierarchy of classes and we can combine a number of related classes together. This can lead to a more reasonable experimental setup. This experiment was set up as a canine vs feline classification problem with each class consisting of a number of subclasses. These subclasses were "foxes", "wolves", "collies", and "swiss mountain dogs" (sennenhunde) for the canine superclass and "domestic cats", "wild cats" and "big cats" for the feline superclass. We cropped the regions consisting of these classes and split the data when there was more than one animal in the picture. The details of the dataset can be found in the appendix.

We used very similar networks as the ones in the experiment with the CIFAR-10 dataset. We still had to resize the data into 224×224 pixel images because the dataset did not contain images of uniform size. The only difference with the networks used in the CIFAR-10 dataset was that we did not add the fully connected layer to the end of resnet. This is because in this case, even the smallest image in the dataset was large enough that compressing it into a 512-dimensional space could be considered a good compression. We found that the same set of hyperparameters as the ones in CIFAR-10 works the best in this experiment too. The only difference was that the first stage of DEPN training was done in only 10 epochs instead of the previous 130. This is because resnet-18 is trained on ImageNet itself and already works very well on this dataset. The results of these experiments can be seen in Table 3

## 4 DISCUSSION

The results from MNIST and FashionMNIST show that DEPN, in comparison to PDL, achieves comparable accuracy for classification of the superclasses while achieving significantly better accuracy for fine-grained nearest prototype labels (Fine). In this particular example, embedded spaces created by MsDNN-AE outperform the ones by plain MsDNN. Also, comparisons with not using MsDNN and only applying the second stage of training (prototype training) show that MsDNN is responsible for a significant portion of the performance in both coarse and fine-grained accuracy. It also shows that PDL degrades the subclass neighborhoods as shown by the fine-grained nearest neighbor accuracy (F-NN) and especially fine-grained prototype label accuracy (Fine). Note that fine-grained nearest neighbor accuracy can be seen as some form of a soft upper

bound for fine-grained accuracy in general (which can be achieved by having as many prototypes as training samples, turning the model into a nearest-neighbor model).

The results for CIFAR-10 continue to show a large improvement in the fine-grained performance of DEPN compared to PDL. One thing to note is that the decoder of PDL in this experiment was extremely blurry, only showing a very vague shape of the original input image and performing even worse for the prototypes themselves. We used the closest sample to the prototype as a proxy of the prototype itself.

The results for the ImageNet dataset are interesting because the Resnet embedding space is trained on ImageNet itself. This means that the embedding space starts very well separated in terms of subclasses. We found that in this case, using MsDNN-AE will actually slightly degrade the embedding space because the decoder is not pre-trained and the backpropagated loss of the decoder will be detrimental to the fine-grained neighborhoods of the embedding space. But plain MsDNN loss will start the first stage already near the local minimum, so it will not perform any drastic change to the embedding space and will keep it largely intact. Our observation of the training procedure aligned with these theories. Also, as the embedding space itself was already well separated, not using MsDNN at all yielded similar results to the plain MsDNN (no autoencoder). PDL, like before, was detrimental to the fine-grained class neighborhoods of the embedding space.

In conclusion, we have offered DEPN, an interpretable prototype-based neural network that achieves good accuracy and offers demonstrably better explanations compared to the state of the art. It works so by offering the prototype that is most responsible for the decision of the network as an explanation. As the prototype and its associated input were selected in a nearest-neighbor-friendly embedding space, the offered explanation is likely to be similar to the sample being classified as opposed to just being from the same class.

We have demonstrated the validity of our method by performing a number of experiments in which we used datasets with coarse-grained and fine-grained class labels to assess the similarity of the offered explanation to the input sample. This has let us achieve an average of 125% improvement on fine-grained accuracy compared to the state-of-the-art. We believe this work has also shown that, when developing an interpretable method, we should mind the quality of offered explanations especially in cases where direct information is limited.

## 5 ACKNOWLEDGEMENTS

This research was enabled in part by support provided by Compute Ontario (`www.computeontario.ca`), Compute Canada (`www.computecanada.ca`) and Natural Sciences and Engineering Research Council of Canada (`www.nserc-crsng.gc.ca`).

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

# A APPENDIX

## A.1 DETAILS OF THE IMAGENET DATASET

We used used 7 subclasses and 2 superclasses in our curated dataset. ImageNet samples might contain multiple objects. These objects have boundaries which we used to crop and split them if necessary.

Each of our subclasses consisted of smaller subclasses. The actual ImageNet labels for those were:

**Canines:**

1. Foxes: 2315 total - 'n02119789', 'n02119022', 'n02120505', 'n02120079'
2. Wolves: 2199 total - 'n02114712', 'n02114855', 'n02114548', 'n02114367'
3. Collies: 2258 total - 'n02106030', 'n02106166', 'n02105855'
4. Swiss mountain dogs: 2251 total - 'n02108000', 'n02107683', 'n02107574', 'n02107908'

**Felines:**

1. Big Cats: 3542 total - 'n02128925', 'n02129604', 'n02128385', 'n02128757', 'n02129165', 'n02130308'
2. Domestic Cats: 2906 total - 'n02124075', 'n02123394', 'n02123159', 'n02123597', 'n02123045'
3. Wildcats: 1108 total - 'n02125311', 'n02127052'

We used the validation set of ImageNet to create the test set for this experiment. The validation set of this experiment was a 10% random subset of the training set.

## A.2 SELECTING THE BEST $\sigma$

To select the best $\sigma$, we started from a small amount and doubled the value each time until we hit the best possible coarse-grained nearest-neighbor validation result. Figure A.2 shows the results of those experiments. We chose based on the best coarse-grained nearest-neighbor accuracy but fine-grained results rose and fell generally in line with those.

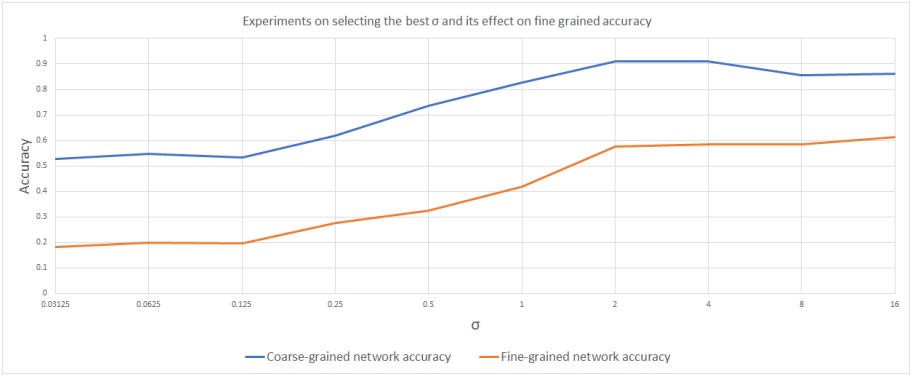

Figure 3: Coarse and fine grained network classification accuracy for different values of $\sigma$ on CIFAR-10.

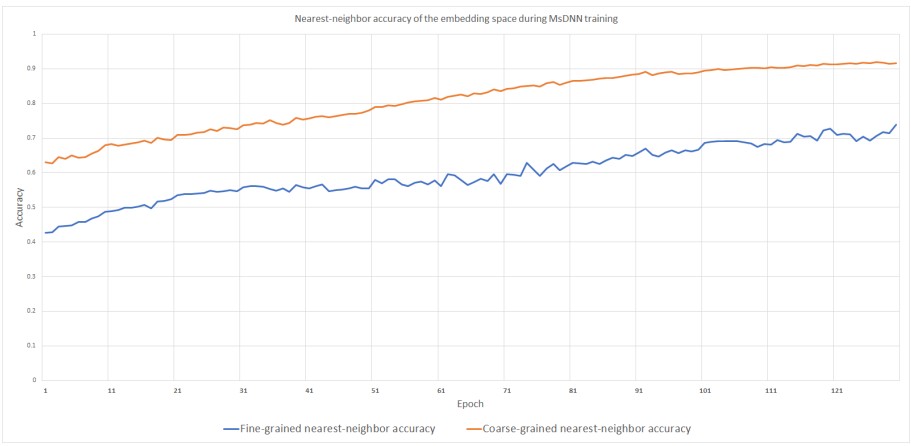

Figure 4: Coarse and fine grained nearest neighbor classification validation accuracy during the first stage of training on CIFAR-10.

## A.3 STOPPING CRITERIA FOR MsDNN AND FINE-GRAINED ACCURACY TRENDS

We stopped training the first stage when the coarse-grained classification accuracy of the network plateaued. Our data shows that fine-grained accuracy followed the same trends as coarse-grained accuracy during the training. Figure A.3 shows one of the training results on CIFAR-10.

## A.4 VARIETY OF THE PROTOTYPES

We found that our method is able to find varied prototypes. Figure A.4 shows an example of them on the ImageNet dataset. Note that the method was able to find prototypes that are not only from the 7 subclasses that we created but also the smaller subclasses that were merged to create them. For example, you can see different types of "big cats" here, including tigers, lions, panthers, etc. An interesting example is that there are two prototypes for lions. One for male and one for female (a distinction that does not exist even in the original ImageNet).

## A.5 SECOND STAGE OF TRAINING

In the second stage of DEPN training, the weights of each individual loss function can have important effects on the final distribution of the prototypes. We found that the weight of the cross-entropy loss should be about the same as the weight of $r_1$. If it is too low, the coarse-grained accuracy of the model might suffer. If it is too high, it will prevent the other loss functions to effectively spread the prototypes.

We found $r_1$ to be more important than $r_2$. If the weight of $r_1$ is too low, prototype diversity and as a result, fine-grained network accuracy suffers. $r_2$ on the other hand, should probably have lower weights than the others. In an ideal situation, it should force the prototypes to basically move to the location of their nearest sample. But our training is done in batches, and while the distance between a prototype and its nearest sample might be 0 in one batch, the same prototype might not be even close to any sample in another. We found that this dependency on batch can be somewhat mitigated by using a larger batch size, but unless the whole dataset is in each batch, this problem remains. We found that if the batch size is too small or the weight of $r_2$ is too large, the prototypes will be pushed to the densest areas of data distribution. That makes sense as those areas are the ones to more likely be represented by at least one sample in each batch. This has a bad effect on

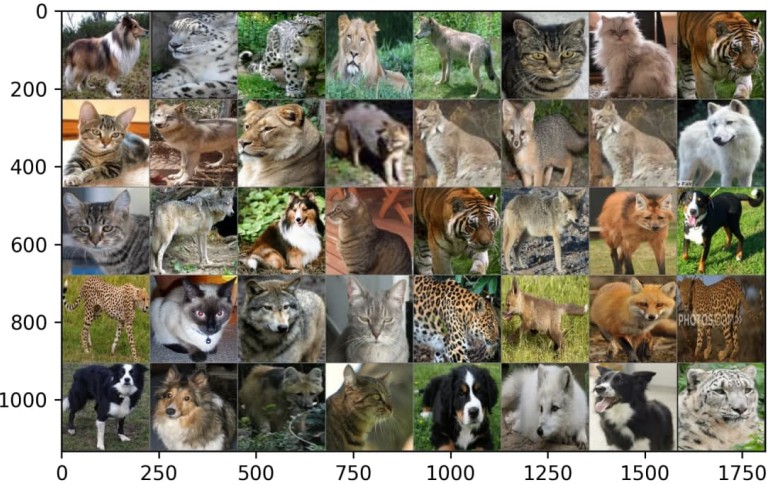

Figure 5: The 40 prototype samples for one of our ImageNet experiments.

prototype diversity, making many prototype samples to be identical. $r_2$ should not be too low though, as that could let the prototypes move outside the distribution if the data distribution is not a convex region.

### A.6 NUMBER OF PROTOTYPES

There are some important aspects to keep in mind for choosing the number of prototypes. If the number is too small, the network will suffer in terms of expressibility and the nearest prototypes will be too far for some samples. If it is too large, it might make it hard for human user to take all of them in mind if they want to explore all of the distances to these prototypes (potential loss of a global view on interpretability). If the number is absurdly large (in the scale of thousands), we might encounter memory and computational problems. The general trend, however, is that the higher the number of prototypes, the closer the performance of the model will get to nearest neighbor classification. Some examples of experiments with different prototype counts (each are the average of 8 experiments with MsDNN's sigma=2. Note that for CIFAR-10, 40 prototypes were used in the experiments mentioned in the paper) can be seen on Table 4. These show a general upward trend for fine-grained accuracy as the number of prototypes increases. Note that coarse-grained network accuracy is still quite similar for all of the experiments.

Table 4: Results of experiments on different number of prototypes for CIFAR-10.

| Number of prototypes | Coarse | Fine |
|---|---|---|
| 5 | 91.50% | 31.02% |
| 30 | 91.14% | 54.85% |
| 40 | 91.12% | 58.29% |
| 50 | 91.65% | 61.00% |
| 100 | 91.83% | 64.16% |

