# OpenReview forum: "Toward Faithful Case-based Reasoning through Learning Prototypes in a Nearest Neighbor-friendly Space."
_ICLR.cc/2022/Conference — ICLR 2022 Poster_

### Official Review · Reviewer_s8S9 · 2021-10-20

**Correctness:** 4
**Technical Novelty And Significance:** 2
**Empirical Novelty And Significance:** 3
**Recommendation:** 6
**Confidence:** 4

**Main Review:**

The paper's main strengths are:
* The combination of nearest-neighbor learning and prototype learning is very natural and appears to work well, empirically. The combination is also well-motivated from a pretty comprehensive literature review. I particularly appreciate the acknowledgment of early prototype work (e.g. Kohonen) which is typically omitted in the recent literature.
* A particularly nice result is that a learned model can also be used as a one-nearest prototype classifier for finer-grained subclasses of the actual classes it was trained on. In other words: the proposed approach learns hierarchical structure that was not explicitly provided in the training labels. As far as I know, this result is a novelty in the literature and motivates the use of prototype/nearest neighbor approaches.
* The paper provides reasonable ablation studies and close baselines, which enables the reader to tease apart the effect of most architectural choices. Similarly, the discussion of results is detailed and illuminating.
* I found the paper easy to follow and clear, with each claim being justified both conceptually as well as empirically.

The paper's main weaknesses are:
* It seems to me that the architecture of the last 2 layers is needlessly complicated. While this architecture is in Line with Li et al. (2018), a more common option (starting from Kohonen, 1995) is to assign a fixed label to each prototype and classifying data by assigning the label of the closest prototype (as the paper does for the fine-grained classification). Typical loss functions include the GLVQ loss from [Sato and Yamada (1995)](https://proceedings.neurips.cc/paper/1113-generalized-learning-vector-quantization.pdf) or variants of the [LMNN loss (2009)](https://www.jmlr.org/papers/volume10/weinberger09a/weinberger09a.pdf). Such an architecture would also be more in line with MsDNN.
Instead, the present paper selects a pseudo-label for each prototype based on the closest data point and proposes new loss components. Why is that? Does it perform better? If so, I would appreciate another ablation baseline that drops the final, fully connected layer and, instead, uses only the distances to the prototypes for classification.
* While the combination of MsDNN and prototype learning is very natural and makes a lot of sense, intuitively, it would have been nice to see a more formal, theoretical justification why this combination improves performance.
* The experimental evaluation includes very reasonable baselines. However, more baselines would have been possible, taking more recent work into account, such as Chen et al. (2019) or [Hase et al. (2019)](https://ojs.aaai.org/index.php/HCOMP/article/view/5265).
* The experimental hyper-parameter settings on page 6 appear ad hoc and are not justified.

Overall, my impression is that the paper is not quite ready, yet. I believe that more experiments and theoretical analysis are needed to fully develop the proposed idea. I do appreciate the core idea, though, and believe that a future iteration of the paper may well be a good contribution to the field.

Beyond the major points mentioned above, there are a few minor recommendations:

* on page 4, the meaning of the hyperparameter $\sigma$ should be quickly explained.
* Please provide standard deviation (or another measure of variance) in Tables 1-3.

**Summary Of The Paper:**

The paper proposes a novel combination of prototype learning and deep nearest neighbor learning in order to achieve an embedding of the input data that is more friendly for prototype learning while maintaining the computational efficiency and intepretability of a prototype approach. In more detail, the approach first trains a Multi-scale deep nearest neighbor network to embed the input data. Then, it jointly learns a prototype layer and a fully connected layer which outputs class logits based on distances to prototypes. Finally, the learned prototypes are replaced by the closest sample from the actual training data to enhance interpretability. In experiments on MNIST, FashionMNIST, CIFAR-10, and ImageNet, the paper shows that the proposed approach outperforms state-of-the-art prototype learning.

**Summary Of The Review:**

Overall, my impression is that the paper is not quite ready, yet. I believe that more experiments and theoretical analysis are needed to fully develop the proposed idea. I do appreciate the core idea, though, and believe that a future iteration of the paper may well be a good contribution to the field.

/edit The authors have resolved the most crucial points in their response. Provided that these crucial arguments are amended to the paper, I believe that it can be published.

---

> ### Author Response · Authors · 2021-11-19
> **Thank you for your reviewing our paper**
>
> Dear Reviewer,
> Thank you for reviewing our paper. It is encouraging that you consider our work "novel", "easy to follow and clear", literature review "comprehensive", ablation studies "reasonable ", "very reasonable baselines", discussion of results "detailed and illuminating" and claims "justified". Please find our responses to your comments below:
>
> Regarding LVQ and other classical prototype learning procedures versus the prototype layer: There are two advantages that the prototype layer has over LVQ and its descendent methods:
>
> 1: It doesn't set the number of prototypes on a per-class basis. This lets the method to assign prototypes to each class based on clusters and sample density rather than fixing a prototype for a specific class. Take a hypothetical scenario where class-A and class-B consist of the same number of samples each, but class-A is concentrated in a single very dense cluster while class-B consists of 10 dense but disjoint clusters. In LVQ, we would need 20 prototypes (assuming equal prototype assignment per class) to properly model this and it would still break the class-A cluster into rather small and arbitrary parts. In our approach, 11 prototypes would be enough.
>
> 2: In our approach prototypes distances have weights. This means that some prototypes can exert a larger "area of influence" over others when classifying samples. This lets some prototypes to be located further inside the main class distribution and more centered on the dense clusters rather than being forced to get to the borders for the sake of better classification boundary.
>
> We performed an experiment on CIFAR-10 and these two differences do indeed lead to noticeable differences in fine-grained accuracy. DEPN achieved ~56\% fine grained accuracy while GLVQ achieved only ~33\% on the same MsDNN trained backbone network. Coarse grained accuracy on the other hand was almost the same.
>
> Regarding comparison with Chen et al. (2019) and Hase et al. (2019):
>
> Chen et al. (2019) creates very different types of prototypes than our method. That makes direct comparisons of prototype quality quite unusual (The way to do so would be to offer the same explanations to humans and see which ones are more helpful, ie. human experiments). The main drawback of this work is that it is limited to image (and with some changes, text) data and cannot be properly implemented over tabular data. We do not claim to offer better or more user-friendly explanations than this work in image classification domain. Instead we offer our method as a more general one that works for different dataset types.
>
> The same can be said about Hase et al. (2019). This work requires hierarchical labels to be present in the dataset and utilizes that to create hierarchical prototypes. DEPN does not use such data and we cannot claim it to be competitive with this method when such hierarchical labels are present. Instead, we offer a more general approach which doesn't need such hierarchical labels for training.
>
> Regarding hyperparameters on page 6: The hyperparameters for Li et al.(2018) come either from that paper itself, or are set so that both networks use similar hyper parameters. Number of prototypes doesn't affect the results by a large amount (see the response to reviewer 2 for more information), the number of epochs were determined based on when validation loss plateaued for all experiments and the choices of $\sigma$, weight for r1 and weight for r2 were discussed in the supplementary materials (weight for r1 was chosen the same as CE due to its importance in positioning the prototypes and that of r2's was intentionally picked smaller so that the relatively small batch size did not lead to the collapse of the prototypes in the densest regions of the distributions) . Network structure was also mostly based on Resnet18.
>
> Regarding defining $\sigma$ of the MsDNN method: Thanks for your suggestion. We will add it in the final version of the paper.
>
> Regarding variance bounds: All experiments are already repeated eight times. We will gladly add standard deviations to the final version of the paper.

---

> > ### Comment · Reviewer_s8S9 · 2021-11-21
> > **Remaining points**
> >
> > Thank you very much for your response. I particularly appreciate the comparison to GLVQ and the explanation of hyperparameter choices. I have two remaining points:
> >
> > * Would it be possible to add the explanation of hyperparameters to the paper, as well?
> > * I'm not 100% certain that I fully understand the difference to Chen et al. and Hase et al., so let me try to verify: Chen et al.'s approach only works on image data and that's the difference? And Hase et al. only works on hierarchical classifications?

---

> > > ### Author Response · Authors · 2021-11-21
> > > **Thank you for your response**
> > >
> > > Thank you for your response. Regarding your points:
> > >
> > > * We will add the explanations to the paper before the discussion period deadline.
> > > * You are correct on both points, but there is more into the case with Chen et al. (2019). The prototypes there are not even directly comparable to ours due to their nature. Any comparison would have to be about which type of prototypes help humans understand the decision better (human experiments) and we expect that the answer would be application-specific. We still do not claim that our prototypes offer a better understanding than Chen et al, but our method is applicable to a larger variety of problems compared to them. Chen et al is limited to only convolutional neural layers (so only images and maybe text data) and is not applicable to for example, fully connected layers.

---

> > > > ### Comment · Reviewer_s8S9 · 2021-11-21
> > > > **Thanks for the clarification**
> > > >
> > > > Thank you for the clarification. I have raised my score accordingly.

---

### Official Review · Reviewer_MCMJ · 2021-10-31

**Correctness:** 4
**Technical Novelty And Significance:** 3
**Empirical Novelty And Significance:** 3
**Recommendation:** 8
**Confidence:** 4

**Main Review:**

The paper addresses an interesting problems which is interpretability of the classification methods that can have various use cases and can demystify the neural net black box for technical and non-technical individuals. It is written well and is easy to follow. The method is positioned well in the existing body of works and the contributions of the work is clear. Experimental settings and results are throughly and very well discussed which makes the work repeatable. Authors have also done a good job discussing the experimental results and sharing the interesting observations.

A couple of recommendations:
- One of the main contributions of the paper that should get credit for it's superior fine-grained performance is the loss function introduced in equation 1. However, it's not clear to what extent r1 and r2 are contributing to the overall performance of the method and which one is more important. From the explanations at page 6, it looks like that the weight of the cross entropy component of the loss function is significantly higher than r1 and r2 weights which alludes to the fact that r1/2 play a less significant role in the method. It would have been good if the authors could have added one or more ablated versions to the experiments to check this.
- Based on the experimental results, C-NN and F-NN mostly outperform Coarse and Fine which makes the reader wonder if instead of classification nearest neighbor search in the embedding space should have been suggested. If the concern is computational costs, it would have been interesting to see the performance of C-NN and F-NN by using approximate nearest neighbor search (e.g. using LSH or k-d trees).

- Minor comments:
-- P3: "Training stops when accuracy using nearest-neighbor classification on a validation set is minimized"
If I understand correctly, training stops when classification error is minimized not the accuracy. Please check.
-- Some hyper parameters seem to be set arbitrarily (e.g. # prototypes = 30) and it's not clear how assigning a different value could have affected the accuracy of the proposed method or how authors have come up with the used values.

**Summary Of The Paper:**

The paper introduces a novel an interpretable neural network for classification tasks that works by assigning input cases to prototypical cases based on their euclidean distance in an embedding space learned by the network. This is achieved by minimizing a loss function consisting of three components: cross entropy loss; distance between each case and the prototype with the same class, distance between each prototype and the closest case to it. The proposed method is specially useful when classification at fine-grained level is performed where in addition to the class to which an instance belongs, predicting the subclass is desirable as well. Experimental results are provided in the paper that validates the merits of the method compared to a set of variations of the proposed method and a state of the art approach as well.

**Summary Of The Review:**

Overall an interesting read and a well written paper. My recommendation is to accept the paper.

---

> ### Author Response · Authors · 2021-11-19
> **Thank you very much for your review**
>
> Thank you very much for your positive review and your valuable feedback! While we are happy that you consider our work "novel", addressing an "interesting problem", "useful", "interesting read", "well written" with "clear contributions" and results that are "very well discussed" and are "repeatable", you also provided a number of critical comments which we respond below.
>
> Regarding r1 and r2: We had done early experiments without them and the results were disappointing. It was mostly because training the position of the prototypes using only CE loss will lead it to position the prototypes for the best possible decision boundary. This would position the prototypes on the border between the class distributions and sometimes even outside of them. The effects of r1 and r2 are discussed in the supplementary materials of the paper.
>
> Regarding using Nearest Neighbor itself instead of the prototype layer: Indeed the issue we had was the computational cost of inference. We tested kd-tree for CIFAR-10 and saw almost identical accuracy to normal nearest neighbors. Nonetheless, inference using kd-tree was still noticeably slower than using our method even when we ran our neural net over CPU. This gap would only widen with larger datasets.
>
> There are also other problems that could arise with nearest neighbors. Large datasets would require large storage capacity on the inference device. Privacy concerns around distributing data could also be an issue: obtaining approval for distributing a handful of data samples is probably more feasible than obtaining it for all of them.
>
> Still, if the dataset is small enough, and there are no privacy concerns and storage limitations, or inference time is not an issue, it is indeed preferred to use nearest neighbors over the prototype layer.
>
> Regarding the minimization of accuracy: You are right. Thank you for mentioning this typo. It was supposed to be maximized, not minimized. It will be fixed in the final version of the paper.
>
> Regarding the number of prototypes: They are somewhat arbitrary, but the choice follows this logic: too few and the network will have suffer in terms of expressibility and also nearest prototypes will be too far for some samples. Too large (in the scale of hundreds) and it would be too hard for a human to take all of them in mind if they want to explore all of the distances to these prototypes (potential loss of a global view on interpretability). Still as we mentioned in the paper, if the number increases, the results would get closer to nearest neighbor classification. Some examples of experiments with different prototypes (each are the average of 8 experiments with MsDNN's sigma=2. Note that for CIFAR-10, 40 prototypes were used in the experiments mentioned in the paper):
>
> Number of prototypes \| 5 \| 30 \| 40 \| 50 \| 100
> ___
> Coarse Accuracy \| 91.50 \| 91.14 \| 91.12 \| 91.65 \| 91.83
> ___
> Fine Accuracy \| 31.02 \| 54.85 \| 58.29 \| 61.00 \| 64.16
>
> As can be seen, there is a general trend where fine-grained accuracy increases with prototype numbers, but not in a drastic manner unless the number of prototypes becomes too small.

---

> > ### Comment · Reviewer_MCMJ · 2021-11-23
> > **Comment on authors' responses**
> >
> > Thanks for your responses, fixing the typo, explaining your kd-tree experiments and your experimental results without r1/r2 in the loss function, and the rationale behind the number of prototypes. Your responses look good to me.

---

### Official Review · Reviewer_NPUz · 2021-11-02

**Correctness:** 3
**Technical Novelty And Significance:** 2
**Empirical Novelty And Significance:** 1
**Recommendation:** 6
**Confidence:** 3

**Main Review:**

I am not expert in deep learning, but I know that prototype method was very
popular in statistical learning. Many commonly used algorithms, such as KNN and
K-means, are exploiting this idea.

The proposed method adapts the idea of prototype-based classification to deep
neural networks, and uses the automatically extracted prototypes to increase the
explainability of neural nets. The proposed algorithm seems work well on MNIST
and ImageNet benchmarks.

My main concern is the novelty of this work, and its lack of comparison to
previous work in similar direction. For example, the DkNN method mentioned in
related works persues a very similar idea like this work, so I think the authors
should compare this work to it. Moreover, the main contribution of this work is
it extends MsDNN with a prototype layer. Again, I'm not familiar with this area,
so I'm not sure whether this improvement is significant or just incremental.

Other than the above problem, I wonder how difficult it is to apply the proposed
network on real problems. The authors have state that it should be applied on
classification problems that each class consists of many subclasses. In order
to show this idea, the authors manually create superclasses in MNIST and
ImageNet dataset. I understand that using this method can verify if the neural
net preserves the subclass structure in its embedding layer. Nevertheless, in
real problem we usually do not have such type of information, so I wonder how
will the proposed method perform on the original MNIST and ImageNet tasks
comparing to more related works.

**Summary Of The Paper:**

This paper presents a method to learn neural nets that preserves the subclass
similarity in the embedding space. The proposed method adds a prototype layer,
which stores the representative prototypes of some subclasses, then a fully
connected MLP is used to output the class label based on examples' distance to
the prototypes.

**Summary Of The Review:**

The novelty of this work could be limited, and the experiments seem not enough
for validating the effectiveness of the proposed approach, so I recommend to reject.
=======================
After reading other reviews and the authors' clarification, I have increased my rating.

---

> ### Author Response · Authors · 2021-11-19
> **Thank you for reviewing our paper**
>
> Dear Reviewer,
>
> Thank you for reviewing our paper. Please find our responses to your points below:
>
> Regarding comparison with DkNN: DkNN is a method used for explaining the decisions of already trained networks. The main idea behind it is this:
>
> * If two samples are near each other in the final layer of the embedding space before the output layer, they should also be near each other in the previous layers.
>
> * This means that if during the inference, the neighbors of an input sample change drastically in the last embedding layer, then there is something wrong with this input sample (eg. it is an adversarial attack to fool our network, or it is one of the edge cases where our network is going to make wrong decisions)
>
> * So, in DkNN, each new sample is scored by how much it's neighbors change as it moves further along the layers of the network. Using this score, they offer some measure of "confidence" for the decision taken by the network itself.
>
> As can be seen here, DkNN is not a training method or network architecture. It is in fact, designed to be independent of the underlying architecture itself. It provides a confidence score for the decision of an already trained network. This is in contrast to the proposed DEPN, which is a particular network architecture and training procedure designed for classification tasks. They are orthogonal methods. In fact, DkNN can be used in conjunction with DEPN to give a confidence score to it's decisions. This complete difference in how they work and the problem they are trying to solve makes comparisons between the two meaningless.
>
> Regarding real world usage: While it is true that we do not have subclass information in the real-world, it is important to note that our method does not need/use subclasses labels. Subclass labels are used only for calculating subclass accuracy which we reported as empirical evidence for the claim that "Our method creates a better embedding space with more faithful nearest neighborhoods than the state of art (Li et al. 2018)". In a real-world problem, we only train our method for superclass (coarse grained) accuracy (which is the same thing done in the experiments provided in this paper). We don't need/use subclass label for training.

---

> > ### Comment · Reviewer_NPUz · 2021-11-23
> > **RE: Thank you for reviewing our paper**
> >
> > Thank you for your clarification. After reading the other reviews and your reply, I have increased my rating.

---

### Author Response · Authors · 2021-11-19
**Summary and Thanks for Reviewer Comments**

We would like to thank the reviewers for their careful reading of the paper and their constructive remarks. In the following we first highlight some common positive points of the reviewers.

Positive points:

**P1: Novelty** "introduces a novel" (R2); "clear contributions" (R2); "proposes a novel combination" (R3); "result is a novelty" (R3); "appreciate the core idea" (R3); .

**P2: Experiments** "thoroughly and very well discussed" (R2); "detailed and illuminating" (R3); "repeatable" (R2); "proposed algorithm seems work well" (R1); "appears to work well, empirically" (R3); "reasonable ablation studies" (R3); "very reasonable baselines" R3; "validates the merits of the method" (R2); "each claim being justified both conceptually as well as empirically" (R3).

**P3: Research implications** "addresses an interesting problems" (R2); "can have various use cases" (R2); "motivates the use of prototype/nearest neighbor approaches" (R3);

**P4: Literature Review** "method is positioned well in the existing body of works" (R2); "a pretty comprehensive literature review" (R3).

**P5: Readability** "interesting read and a well written paper" (R2); "easy to follow and clear" (R3);

For questions and concerns, please find our response to to each reviewer.

---

### Author Response · Authors · 2021-11-21
**Revision info**

A new revised version of the paper has been uploaded. The changes are:

a) A typo has been fixed on Page 3 (Training stops when accuracy using nearest-neighbor classification on a validation set is minimized. -> Training stops when accuracy using nearest-neighbor classification on a validation set is maximized.) (R2)

b) 95% confidence bounds are added to the result tables (R3)

c) The parameter $\sigma$ is explained on Page 4 (R3)

d) Added more explanations for the choice of hyperparameters to the paper on Page 6 (R3)

e) Added the results of the experiments with different number of prototypes to the appendix (R2)

---

### Decision · Program_Chairs · 2022-01-20

**Decision:**

Accept (Poster)

**Comment:**

The authors propose a neural network model to preserve the sub-class similarity. The key of the model is to add a prototype layer to a multi-scale deep nearest neighbor network. The prototype layer stores the representative prototypes of some fine-grained sub-classes. The use of the prototype layer preserves intepretability and computational efficiency. Experimental results demonstrate that the proposed approach reaches state-of-the-art prototype learning performance.

The reviewers generally find the paper clear and with sufficient contributions. The empirical validation is sufficiently thorough to back the claims in the paper. The main concern prior to the rebuttal among some of the reviewers was about the novelty of the paper (e.g. with respect to DkNN), but the authors convinced most of the reviewers in the rebuttal about the key differences. The authors are encouraged to highlight the novelty aspect more clearly in the revision. Another suggestion was to add an ablation study to justify the importance of the r1/r2 parameters, and the authors have done a successful job addressing the suggestion. Several other comments, such as explanations of the hyperparameters, have been taken into account in the revision. The reviewers thus reach the consensus to recommend acceptance.